# Pathogen Biocontrol Using Plant Growth-Promoting Bacteria (PGPR): Role of Bacterial Diversity

**DOI:** 10.3390/microorganisms9091988

**Published:** 2021-09-18

**Authors:** Hao Wang, Runjin Liu, Ming Pei You, Martin J. Barbetti, Yinglong Chen

**Affiliations:** 1State Key Laboratory of Soil Erosion and Dryland Farming on the Loess Plateau, Research Center of Soil and Water Conservation and Ecological Environment, Chinese Academy of Sciences, Xianyang 712100, China; hao_wang@nwafu.edu.cn; 2University of Chinese Academy of Sciences, Beijing 100049, China; 3Institute of Mycorrhizal Biotechnology, Qingdao Agricultural University, Qingdao 266109, China; liurj@qau.edu.cn; 4The UWA Institute of Agriculture, and School of Agriculture and Environment, The University of Western Australia, LB 5005, Perth, WA 6009, Australia; mingpei.you@uwa.edu.au (M.P.Y.); martin.barbetti@uwa.edu.au (M.J.B.)

**Keywords:** rhizosphere, bacterial mixture, compatible strains, plant defense, disease control, sustainable agriculture

## Abstract

A vast microbial community inhabits in the rhizosphere, among which, specialized bacteria known as Plant Growth-Promoting Rhizobacteria (PGPR) confer benefits to host plants including growth promotion and disease suppression. PGPR taxa vary in the ways whereby they curtail the negative effects of invading plant pathogens. However, a cumulative or synergistic effect does not always ensue when a bacterial consortium is used. In this review, we reassess the disease-suppressive mechanisms of PGPR and present explanations and illustrations for functional diversity and/or stability among PGPR taxa regarding these mechanisms. We also provide evidence of benefits when PGPR mixtures, rather than individuals, are used for protecting crops from various diseases, and underscore the critical determinant factors for successful use of PGPR mixtures. Then, we evaluate the challenges of and limitations to achieving the desired outcomes from strain/species-rich bacterial assemblages, particularly in relation to their role for plant disease management. In addition, towards locating additive or synergistic outcomes, we highlight why and how the benefits conferred need to be categorized and quantified when different strains/species of PGPR are used in combinations. Finally, we highlight the critical approaches needed for developing PGPR mixtures with improved efficacy and stability as biocontrols for utilization in agricultural fields.

## 1. Introduction

In the rhizosphere micro-habitat, plants continuously interact with a plethora of microbes, including bacteria, fungi and viruses [1]. Interactions between beneficial microbes and pathogens are of great significance in plant health and growth and thus have received substantial attention [2,3,4]. However, most research has focused solely on the relationship between a single pair of interacting species, (i.e., one pathogen and one antagonist), ignoring the immense microbial diversities within these functional communities coexisting around or in plant roots. Hence, such studies largely do not relate to natural soil conditions and are inconsistent with the view that diverse species operate in microbial communities [5].

Plant Growth-Promoting Rhizobacteria (PGPR) represent a diverse category of microbes associated with many plant species and bringing benefits to plants, such as growth promotion and stress alleviation. There is a large body of literature demonstrating the potential use of PGPR as biological control agents and for replacing chemical fertilizers and pesticides/fungicides [6,7,8]. The most commonly studied bacteria in relation to biocontrol are members of the genera *Pseudomonas*, *Bacillus*, *Azospirillum*, and *Streptomyces* [2,8,9]. Multiple mechanisms of action have been postulated as to how this defense arises, including direct competition for nutrients and niches, antibiosis, enzyme lysis, signal interference and indirect induction of host resistance [7,8,10,11,12]. The accumulated evidence suggests PGPR taxa vary in the expression of traits [13,14,15] that correlate with one or more of the above mechanisms, thereby altering their capacities for protecting host plant from pathogens and thus promoting plant growth. Since PGPR live in communities on the root surface or sometimes inside the root (i.e., endophytes) and they are recruited by host plants, it has been deemed that each bacterial ‘component’ offers specific benefits for plants [16]. Therefore, there is a critical need to better understand such situations in terms of multi-species PGPR assemblages to maximize desired benefits to plants, particularly under the conditions of pathogen challenges. This review focuses on critical issues concerning PGPR-mediated disease suppression, especially the mechanisms of disease suppression by PGPR mixtures, the efficacy and limitations of PGPR mixtures on biological control, and the determinant factors for successful use of different species/strains of PGPR in mixtures.

## 2. Mechanisms of PGPR in Disease Suppression

### 2.1. Competitive Rhizosphere Colonization

The variable and inadequate biocontrol of PGPR in field tests has usually been correlated with their poor rhizosphere colonization [17,18]. Bacterial inoculants must establish themselves in the plant rhizosphere at population density levels sufficient and persistent enough to generate plant beneficial effects, such as disease suppression. This is dependent upon the ability of bacterial inoculants to proliferate and efficiently colonize the root system, which is well-recognized as the limiting step for biocontrol [19]. Plants exude organic nutrients to the root surface and rhizosphere, among which various nutrients provide niches attracting a diverse range of microorganisms, including pathogens (Figure 1). The common metabolites from root exudates consist of sugars, amino acids, organic acids, vitamins, nucleosides, phytosiderophores, and phenolic compounds [20], and simultaneously, they function as chemical signals for motile bacteria to migrate towards the root surface [21]. PGPR are differentiated to sense these chemo-attractants during root colonization. Examples include the chemotactic responses of *Corynebacterium flavescens* and *Bacillus pumilus* induced by rice root exudates, which were stronger than those of other bacteria [22], and *Azospirillum* strains that show varying degrees of attraction and/or chemotaxis to organic acids, sugars, and amino acids [23]. Recently de Souza et al., (2019) using comparative genomic analysis, found that growth-promoting traits of PGPR involving carbon and nitrogen acquisition, rather than other traits such as auxin production, are deterministic for their successful colonization and population establishment in the rhizosphere [24]. The components and quantities of compounds exuded by plant roots are determined by the genetic constitution of particular plant types and by environmental factors [25]. This implies that PGPR colonization and/or competence is not only closely connected with their capacities to avail of a specific environment and to adapt to changing conditions, but also connected with their capacities to operate as diversified nutrient uptake phenotypes in PGPR.

Nutrients and physical sites are crucial for the establishment of populations of epiphytic microorganisms, including pathogens and nonpathogens alike. Competition for nutrients and for occupation of niches at root surfaces is an indirect but important antagonism by PGPR against pathogens that depend on such external resources [26]. For example, in the rhizospheres of different crops, certain bacterial strains inhibit oospore germination of *Pythium aphanidermatum* by competing for glucose and asparagine [27], and *Pseudomonas fluorescens* PJ0210 competes for glucose with the pathogen *Bipolaris maydis*, the causal agent of leaf blight in corn [28]. An extraordinary form of nutrient competition is limitation by iron. Normally in aerobic soil, iron is present in insoluble forms (i.e., III) that are barely or non-accessible for many living organisms. However, under conditions of iron deficiency, PGPR have evolved to acquire ferric iron through the production of low-molecular-weight compounds known as siderophores [29], allowing solubilization of iron and its access from mineral or organic complexes. Thus, siderophore production by PGPR gives them competitive advantages in colonizing roots and in excluding pathogenic microbes from rhizosphere ecological sites [30]. Different categories of bacterial siderophores have been identified and mainly include hydroxamates, carboxylates, and catecholates [31], and they show varied abilities to sequestrate iron *in vitro*. In general, they have higher affinity for ferric iron, particularly those produced by fluorescent *Pseudomonas*, compared to the fungal siderophores [29,32]. Deprivation of ferric iron by *P. putida*-producing siderophores mediates suppressiveness to Fusarium wilt pathogens of cucumber, radish, and flax [32]. Fluorescent *Pseudomonas* EM85 and *Bacillus* spp. [MR-11(2) and MRF] isolated from maize rhizosphere, produce siderophores that suppress root rot disease [33]. Siderophore production by *P. fluorescens* 3551 or *P. putida* N1R contributes to its antagonistic activity against *P. ultimum* [34,35]. Similarly, fluorescent *Pseudomonas* secreting the siderophore pyoverdine induces suppression of the fungal pathogen *Botrytis cinerea* by depleting iron [36]. Sneh et al. (1984) demonstrated that both chlamydospore germination and mycelia growth of *Fusarium oxysporum* were suppressed more by siderophore-producing *P. fluorescens* than other isolates [37]. Some *Pseudomonas* strains are even able to utilize heterologous siderophores secreted by root-colonizing pathogenic microorganisms [38,39].

Mutants defective in motility are significantly less competitive for root colonization and therefore not capable of controlling fungal root pathogens, and vice versa [40,41]. Additionally, other studies have indicated that biofilm formation is also a determinant in rhizosphere colonization by PGPR, such as where mutants unable to synthesize exopolysaccharide were unable to form biofilms and efficiently colonize the rhizosphere, and thus rendering low population levels attached to the rhizosphere [42,43]. Moreover, using RNA sequencing, Guo et al. (2020) proved that *Streptomyces pactum* Act12 inoculation enhanced tomato rhizosphere colonization and competition by indigenous *P*. *koreensis* GS via the promotion of swimming motility, biofilm formation, and environmental adaptation [44].

From a microorganism perspective, the nutrient niches in the rhizosphere soil are frequently limited. Increasing the richness of PGPR taxa (and thus functional diversity) colonizing the rhizosphere, and their ability to do so across a wider range of biotic and abiotic conditions, would likely ensure greater rhizosphere competence. In addition, importantly, it should also improve their ability to outcompete pathogens for limited resources available and so making them unavailable for pathogenic microbes to acquire and develop.

### 2.2. Antibiosis

Toxins that kill other organisms at low concentrations (<10 ppm) are termed antibiotics [45]. Antibiotics comprise a heterogeneous group of low molecular weight, organic compounds [46], which interfere with the synthesis of pathogen cell walls, cell membrane structures, and the biogenesis of initiation complexes on the smaller subunit of the ribosome [47]. Accumulated reports on antibiotic-mediated pathogen suppression mainly include *Pseudomonas* spp. producing 2,4-diacetylphloroglucinol (DAPG) [48,49,50,51], phenazine ([52,53,54], pyoluteorin [55,56,57], pyrrolnitrin [58,59,60], hydrogen cyanide (HCN) [61,62], and *Bacillus* spp. producing bacillomycin D [63,64], mycosubtilin [65,66,67], and other lipopeptides including iturin A, fengycin, and surfactin [68,69,70]. In addition, it has also been observed that xanthobaccin A secreted by *Stenotrophomonas* sp. strain SB-K88 [71], zwittermicin A [72,73], and kanosamine [74] produced by *Bacillus cereus* UW85, prodigiosin produced by *Serratia* spp. [75,76], and volatile organic compounds like ketones and pyrazine released by *Bacillus* spp. [69], can all be antagonistic against pathogenic fungi *in vitro*.

Howell and Stipanovic (1980) found that antibiotic pyoluteorin synthesized by *P. fluorescens* Pf-5 was inhibitory to *Pythium ultimum*, but not to *Rhizoctonia solani*, on cotton [55]. Nielsen and Sørensen (2003) showed that fluorescent *Pseudomonas* strains differ in their abilities to produce antifungal cyclic lipopeptides (CLPs) in the sugar beet rhizosphere, e.g., *P. fluorescens* DR54 accumulated a high viscosinamide level, while *P. fluorescens* strains 96.578 and DSS73 exhibited a significant accumulation of tensin or amphisin [77]. Moreover, certain *P. fluorescens* strain groups (e.g., biotypes I, V, or VI) are affiliated with the production of specific CLPs against root-pathogenic fungi [78]. Antibiotic synthesis by PGPR closely relates to the cell metabolic status, which, in turn, is influenced by biotic and abiotic stimuli [79], including host plant growth, type of substrate and supply, temperature, oxygen availability, and pH. It is noteworthy that many strains can produce suites of secondary antibiotics, and that conditions favorable for production of one antibiotic compound are likely unfavorable to produce another [80]. This being the case, importantly, the assemblages of diverse PGPR strains offer a higher degree of flexibility and biocontrol effectiveness for the antagonists, particularly when PGPR are confronted with heterogeneous or changeable conditions.

### 2.3. Enzyme Lysis

A wide variety of PGPR showing hyperparasitic activity attack pathogens through secretion of cell wall hydrolases, such as chitinases, glucanases, cellulases, and proteases [81]. For example, *Serratia plymuthica* secreting chitinase hampers germination and germ tube elongation of the pathogenic fungus *Botrytis cinerea* [82]. The ability to produce β-1,3-glucanase is critical for *Pseudomonas cepacia* to deconstruct cell walls of *Rhizoctonia solani*, *Pythium ultimum*, and *Sclerotium rolfsii* [83]. Singh et al. (1999) showed that *Paenibacillus* sp. 300 and *Streptomyces* sp. 385 both synthesize chitinase and β-1,3-glucanase to lyse cell walls of the pathogenic fungus *Fusarium oxysporum* f. sp. *cucumerinum* [84]. *Bacillus* sp. BPR7 possesses chitinase and β-1,4-glucanase activities that inhibit the growth of fungal pathogens, such as *F*. *oxysporum*, *F*. *solani*, and *R*. *solani*
*in vitro* [85]. *Bacillus thuringiensis* UM96 produces chitinase to inhibit mycelial growth of *B. cinerea* [86], the cell walls of which could also be degraded by *B. atrophaeus* JZB120050, which produces chitinase, glucanase, and protease [87]. *Bacillus subtilis* RH5 secretes chitinase, protease, cellulase, and xylanase that act against *R. solani* [88]. Further, *Paenibacillus ehimensis* KWN38 showed high activities of chitinase, cellulase, glucanase, and protease against *R. solani* AG-1, *F. oxysporum* f. sp. *lycopersici*, and *Phytophthora capsici* [89].

Chitinolytic activity seems non-essential for *Serratia plymuthica* IC14 acting against *B. cinerea* and *Sclerotinia sclerotiorum*; rather, it involves the synthesis of other enzymes, such as proteases [90]. Similarly, of the hydrolytic enzymes cellulase, protease, chitinase, and pectinase produced by *Paenibacillus* sp. B2, only protease was responsible for inhibition of the mycelial growth of *Phytophthora parasitica* [91]. Importantly, it is highly likely that a wider array of enzymes produced by different species of PGPR in mixture will have greater advantage of suppressing multiple pathogens present in the host rhizosphere due to complementary action of their lytic enzymes.

### 2.4. Induction of Systemic Resistance

Induced systemic resistance (ISR) is a state of active resistance due to an inducing agent after pathogen infection. ISR can be induced by beneficial rhizobacteria, whereas the pathogen-induced resistance is called systemic acquired resistance (SAR) [92] (Figure 1a). The induced resistance confers non-specific protection against a broad spectrum of attackers, including fungi, bacteria, virus, insects and nematodes [12]. SAR involves salicylic acid (SA) and changes in gene expression related to pathogenesis-related proteins (PR proteins). Most PGPR employ an SA-independent pathway to activate ISR, a pathway involving jasmonate and ethylene signaling [93]. These hormones are implicated in activating certain sets of defense-related gene expression in plants and/or spreading the defense mechanisms into distal plant tissues, leading to host morphological and metabolic responses, such as cell wall strengthening, accumulation of PR proteins or defense-related enzymes (e.g., chitinase, glucanase, peroxidase, polyphenol oxidase, phenylalanine ammonia lyase, chalcone synthase, lipoxygenase, etc.), and syntheses of phenolic compounds and phytoalexins [94,95].

Several bacterial traits have previously been identified as elicitors of ISR, including flagella, cell envelope component lipopolydaccharides (LPS), and secreted metabolites such as siderophores [94,96], salicylic acid [97,98], and antibiotics like 2,4-diacetylphloroglucinol (DAPG) [99] and pyocyanin [100]. The accumulated evidence indicates that many other traits of PGPR are also operative in eliciting ISR. For example, volatile organic compounds 2R,3R-butanediol emitted by *Bacillus subtilis* GB03, branched-chain alcohols by *B. amyloliquefaciens* IN937a [101,102], dimethyl disulfide (DMDS) by *B. cereus* C1L [103], and quorum-sensing molecules including *N*-acylhomoserine lactones (AHLs) produced by *Serratia liquefaciens* MG1, *Pseudomonas putida* IsoF [104], and *Serratia plymuthica* HRO-C48 [105]. Some of the ISR elicitors function redundantly, e.g., the flagella, LPS-containing cell walls, and siderophore pseudobactin of *P. putida* WCS358 all can induce systemic resistance in *Arabidopsis thaliana* when exogenously applied to the roots [106]. However, *P. putida* WCS358 mutants, defective in flagellation, in immunizing O-antigenic side chain of the lipopolysaccharides or in pseudobactin synthesis, were still effective in triggering ISR [106].

Strains of PGPR vary in effectiveness in inducing systemic resistance [107], and the induction of systemic resistance is reliant on strain-specific traits [108]. Importantly, Jetiyanon and Kloepper (2002) showed that mixtures of compatible dual PGPR strains eliciting induced systemic resistance to different pathogens in several plant hosts provided greater suppression of diseases than the individual strains [109]. Similarly, a dual bacterial consortium containing *P. putida* CRN-09 and *B. subtilis* CRN-16 conferred significantly greater expression of ISR to *Macrophomina phaseolina* in mung bean as compared with the application of a single strain, by enhancing activities of phenylalanine ammonia lyase, polyphenol oxidase, peroxidase, β-1,3-glucanase, and chitinase [110]. Dutta et al. (2008) using a split root experiment, found that the combination of RRLJ04 or BS03 with the rhizobial strain RH2 was better in inducing systemic resistance than their individual treatments [111]. Further, Berendsen et al. (2018) demonstrated that while a *Xanthomonas* sp., a *Stenotrophomonas* sp., and a *Microbacterium* sp. did not affect the plant separately, the triple bacterial consortium induced systemic resistance against *Hyaloperonospora arabidopsidis* and promoted plant growth [112]. These studies highlight the increased expression of and benefits from varied traits using PGPR mixtures. It is clear that the use of PGPR mixtures covers greater trait variation and complementarity and hence increases the likelihood of success, and at least reliability, in activating host systemic resistance against pathogen infection and provide broad-spectrum protection against different pathogens.

### 2.5. Signal Interference

Many phytopathogenic bacteria only evoke pathogenicity or virulence factors at a high bacterial population density by detecting the accumulated levels of quorum-sensing (QS) molecules (also known as autoinducers) such as AHLs [11]. Signal interference is a mechanism of biocontrol based on the interference with QS through enzymatic degradation of the autoinducers. For example, lactonases of *Bacillus thuringiensis* hydrolyze the AHL lactone ring [113], and acylases of *Ralstonia* sp. cleave the AHL amide [114]. Growing evidence highlights additional AHL-degrading enzymes with different specificity and kinetics in a great variety of PGPR species/strains, among which are *Bacillus* spp. [115,116,117], *Pseudomonas* spp. [118,119,120], *Comamonas* sp. D1 [121,122], *Arthrobacter* sp. IBN110 [117], *Actinobacter* sp. C1010 [123], *Rhodococcus erythropolis* W2 [121,124], *Streptomyces* sp. M664 [125], *Rhodococcus* sp. LS31 and PI33 [126], *Ochrobactrum* sp. T63 [127], *Mesorhizobium* sp. S5 [116], and *Microbacterium testaceum* StLB037 [128,129]. Jayanna and Umesha (2017) reported that the ability of *P. aeruginosa* 2apa to degrade AHL by acylase also inhibits the biofilm formation of *Ralstonia solanacearum*
*in vivo* [120]. Biocontrol is likely fostered when biofilm formation is impaired or absent. *In Vitro*, *Rhodococcus erythropolis* and *Bacillus simplex* reduced the pathogenicity of *Agrobacterium tumefaciens* [121] and *Erwinia amylovora* [130] via quenching AHL-quorum sensing. Studies are rare that examine the effects of PGPR assemblages of varying richness with regard to this mechanism. However, the wide diversity of AHL-degrading PGPR used in assemblages may provide greater opportunities for effective biocontrol towards AHL-utilizing phytopathogens.

## 3. PGPR Mixtures in Disease Suppression

In natural habitats, PGPR live in multi-species assemblages in soil or plant rhizosphere [20,131]. Given the community-based lifestyle of PGPR, it is advocated to use mixed PGPR of diverse species to enhance the efficiency and reliability of disease control in different agricultural fields, with an assumption that the mixture will confer synergistic control of the tested pathogens (Table 1). *Pseudomonas fluorescens* F113 and *Stenotrophomonas maltophilia* W81 prevent damping-off of sugar beet through the production of DAPG and extracellular proteolytic activity, respectively, and in a field experiment only co-inoculation of W81 and F113 prevented the disease [132]. Both *P. fluorescens* sp. M23 and *Bacillus* sp. MRF produce antifungal antibiotics and siderophores, and are efficient in rhizosphere colonization, such that when co-inoculated on maize plants there was significantly decreased infection of *Fusarium* spp. in comparison with untreated control plants and with a single bacterial agent treatment [33]. Similarly, the mixture of *Bacillus amyloliquefaciens* IN937a and *B*. *pumilus* IN937b elicited systemic resistance, leading to more consistent broad-spectrum pathogen control in various crops under field conditions in comparison with an individual strain [133], and this *Bacillus* strain mixture had 25–30% greater superoxide dismutase and peroxidase activities than the non-bacterized control [134]. In the same way, a combination of *P. putida* strains WCS358 and RE8 reduced Fusarium wilt incidence in radish by up to 50% as compared to the 30% reduction from the individual strain [135]. In this case, by applying the strain mixture, two different disease-suppressive mechanisms (i.e., competing for iron through pseudobactin production for WCS358, and inducing systemic resistance for RE8) acted together to enhance disease suppression. It was also possible that these two strains colonized different niches and so limiting competition between them for iron [135]. *Burkholderia* spp. RHT8 and RTH12 both showed the production of siderophores as well as chitinase and β-1,3-glucanase; and the co-inoculation treatment suppressed *Fusarium oxysporum* leading to increased growth and yield of fenugreek in both *in vitro* and in field conditions, as compared to single inoculation and non-inoculated control [136]. In these examples, enhanced disease suppression in a bacterial mixture not only likely involves different disease-suppressive mechanisms but may also result from interactions between two or more introduced PGPR strains positively affecting (anti-pathogen) activity, root colonization, and growth of the bacterial strains.

## 4. Limitations to PGPR Mixture

There is also evidence that certain PGPR mixtures do not exhibit synergistic or comparable effects on disease control and plant growth, with respect to their single strains [140,141,142]. For example, a mixture of lytic non-fluorescent and siderophore-producing fluorescent bacteria did not increase suppressiveness to Fusarium wilt of cucumber [37]. *Pseudomonas chlororaphis* PCL1391 and *P. fluorescens* WCS365 suppress plant diseases mainly by the production of antibiotic phenazine [53] and by induction of host systemic resistance [61], respectively. However, although the combination of the two bacteria promoted plant growth, there was no significant difference in control of bean anthracnose from the mixtures as compared to only treatment with strain PCL1391 [143].

The antagonism between biocontrol bacterial agents used in mixtures or between a biocontrol agent and the indigenous microflora can undermine the performance of bacterial agents in the rhizosphere. This is particularly so when two or more populations of microbes colonize the same ecological niche and have similar nutritional requirements [144] such that competition for niches and nutrients (niche exclusion) will be inevitable. For example, effective iron competition by endemic *Pseudomonas* spp. led to the ineffective control from *Trichoderma hamatum* for *Pythium* seed rot of pea [145]. Similarly, *P. putida* WCS358 decreased rhizosphere colonization of radish by *P. fluorescens* WCS374 and eight indigenous *Pseudomonas* strains, and siderophore-mediated competition for iron was the main determinant in these negative interactions [146]. Additionally, competition for limited carbon between *P. fluorescens* Ag1 and *Alcaligenes eutrophus* JMP134 decreased the population size of JMP134 in the rhizosphere of barley [147]. Secondary metabolites secreted by one organism impeding the growth of or disease control from the other organism is another antagonism that can occur between two populations of biocontrol agents [148]. For example, Molina et al. (2003) illustrated that *Pseudomonas chlororaphis* PCL1391 suppresses *Fusarium oxysporum*-inducing vascular wilt of tomato by production of phenazine, which is controlled by AHL-mediated QS. When co-inoculated with AHL-degrader *P. fluorescens* P3/pME6863, the antifungal activity and protection of this biocontrol agent against vascular wilt was markedly attenuated [149]. These negative interactions can restrict the activity of, and/or the colonization by, inoculated PGPR strains, particularly where the rhizosphere population density of one or all strains fail to reach the threshold level needed for disease suppression to occur [17,19]. In contrast to the above examples, Felici et al. (2008) found that a lack of synergistic impacts of dual bacterial inoculation (*Bacillus subtilis* 101 and *Azospirillum brasilense* Sp24) was not related to reduced persistence of one or both bacteria in the rhizosphere, but rather due to the implication of independent signaling pathways associated with different modes of action in the two bacterial species [141]. Hence, compatibility of strains of PGPR mixtures is an essential prerequisite for better biocontrol efficacy and stability of biocontrol agents. Further, the interactions (e.g., synergistic, antagonistic, and neutral) between members of synthetic microbial communities shape their functioning and evolution in either constant or in fluctuating environments [150], but historical studies have rarely assessed the fate of bacteria in soil when introduced as a mixture, nor the effect of bacterial mixtures on the microbial communities including macro-organisms, present in the rhizosphere. This critical area of research deserves far more attention in order to better utilize PGPR mixtures in improving their efficacies.

## 5. Critical Approaches towards Developing Successful PGPR Mixtures

Various rhizosphere bacteria are potential biological pesticides capable of protecting plants against diseases and improving plant fitness and yield. To increase and maintain the level of biological control, multiple strain mixtures of PGPR have been employed successfully in many crops, especially when individual strains are unable to provide adequate suppression of pathogens. A range of biocontrol mechanisms, such as competitive rhizosphere colonization, secretion of antibiotics and enzymes, signal interference, and induced systemic resistance, may operate in mixed PGPR populations and strengthen the ability of the combined partners in an additive or synergistic manner (Figure 1), which is possibly correlated to the potency exerted by biodiversity [151,152]. Although the relative significance of each mechanism is unknown and might vary with circumstances, it is evident that multiple mechanisms function in biocontrol systems under field conditions. Clearly, PGPR mixtures have the advantage of combining their diverse traits, in particular the traits that are hard to find in a single bacterium. Moreover, as PGPR mixtures more closely mimic the microbial communities present in the rhizosphere, application of such mixtures should enable better preparation and tolerance when faced with the challenges of varied biotic or abiotic conditions that may otherwise reduce variability in biocontrol efficacies [153]. This increased level of stability in biocontrol is also often observed in mixtures of plant cultivars [154], of fungicides [155], and sometimes even of arbuscular mycorrhizal fungi [156].

Since external biotic and abiotic factors shape the microbial communities in soil [157], the performance of different mechanisms of biocontrol by PGPR is likely dissimilarly affected by them. Dominant factors comprise challenges, such as inadequate rhizosphere colonization, limited tolerance to environmental/climate changes, and fluctuating production or activity of antimicrobial metabolites (antibiotics, enzymes, etc.) [8,158,159], which are often overlooked when PGPR mixtures are developed artificially to treat plants. While artificially combined PGPR mixtures may bring in increased, unchanged, or decreased disease-suppressive effects [160], there remain significant prospects for increased disease control from PGPR mixtures if the underlying interactions are better understood. Several determinant factors presumably account for the success of some PGPR mixtures in disease control. First, individual PGPR strains/species have differing substrate utilization profiles with niche preferences or differentiation in the root zone [161], where higher levels of coexistence or compatibility should be expected, provided that PGPR strains differ in their ecological requirements for survival, colonization, and activity. Additionally, diverse bacterial populations occupy different niches in the rhizosphere and/or generate specialization in the same niches [6], and hence restrict competition from competing strains/species of PGPR and strengthen cooperation among them. Second, individual PGPR strains/species exert complementary disease-suppressive mechanisms (traits) [132,135], such that when one mechanism is ineffective under a particular set of conditions, the others can compensate for the former absence. Third, the effects of similar or different mechanisms of action employed by different strains may augment quantitatively in a beneficial way [44,112] (Figure 1b).

Recently, using metatranscriptomic analysis, Gómez-Godínez et al. (2019) revealed that *Azospirillum nif* genes were upregulated in the presence of other PGPR species, resulting in active nitrogen fixation by *A. brasilense* in maize roots [15]. Similarly, it was shown that the individual bacterial agents within PGPR communities differentially express their disease-suppressive traits [14,16], and accordingly induce the tuning of genes and metabolic pathways in host plants to achieve specific targets that benefit agriculture. Indeed, specific interactions between PGPR strains can influence the level of pathogen suppressiveness through combination of these strains [148,160], and the functioning of individual strains within a bacterial consortium can be used to predict the performance of the bacterial communities and associated phenotypes in the hosts [162]. To secure additive and more synergistic interacting outcomes, future investigations into the use of different strains/species of PGPR in combinations need to quantitatively determine the key biocontrol processes and their interactions, and the benefits conferred should be categorized and quantified (e.g., via functional analyses employing transcriptomics, proteomics, and metabolomics under contrasting conditions). For instance, it would be informative to determine how bacterial mixture-mediated metabolic and transcriptional regulations are positively associated with plant defense responses during biotic and/or abiotic challenges [163]. This represents the next logical step towards the development of compatible PGPR mixtures with improved biocontrol efficacy and stability for utilization in heterogeneous agricultural systems.

## Figures and Tables

**Figure 1 microorganisms-09-01988-f001:**
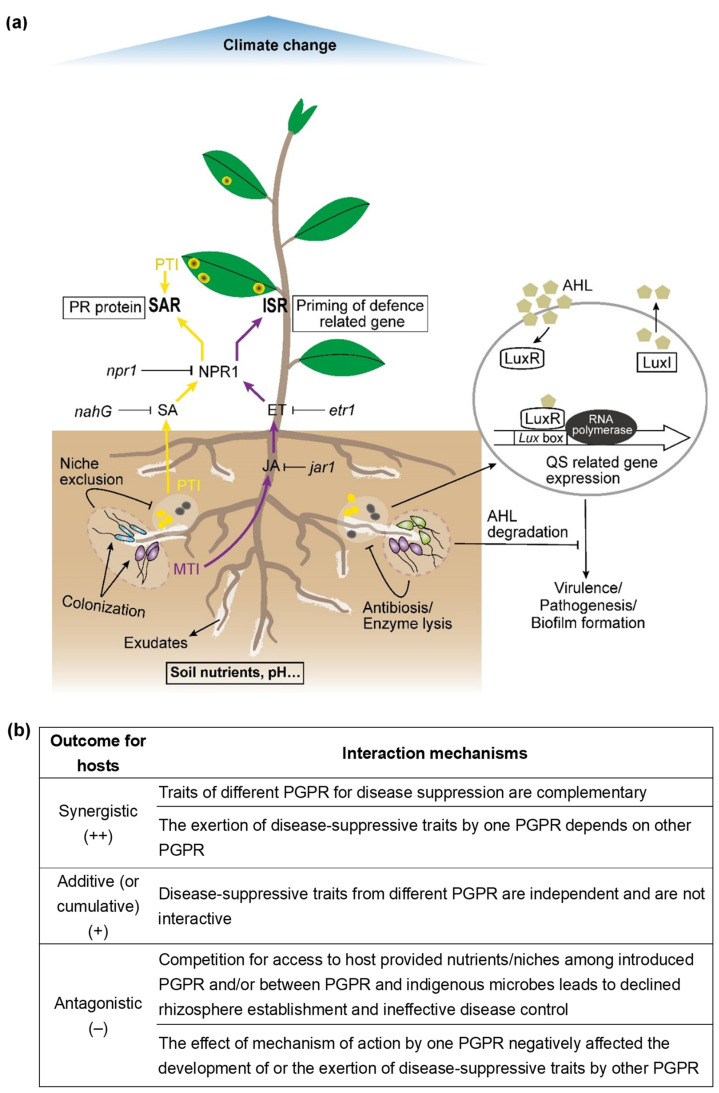
(**a**) The introduced multi-species PGPR assemblages (represented in blue, purple, and light green) and their diverse biocontrol mechanisms in protecting plants against different pathogens (represented in dark grey and yellow or their blend) present in rhizosphere soil or aboveground plant parts. PTI/MTI, pathogen-/microbe-triggered immunity; QS, quorum sensing; AHL, N-acylhomoserine lactone; SA, salicylic acid; JA, jasmonic acid; ET, ethylene; SAR, systemic acquired resistance; ISR, induced systemic resistance; NPR, nonexpresser of pathogenesis-related (PR) genes; LuxI, AHL synthase; LuxR, AHL regulator. (**b**) This table describes a set of interaction mechanisms between PGPR that could create the likely outcomes (synergistic, additive, or antagonistic) for the host plants.

**Table 1 microorganisms-09-01988-t001:** PGPR mixtures effective against different plant pathogens along with their mechanisms of action with example case studies.

Mechanisms of Action	Host Plants	Diseases/Pathogens	Effective PGPR Inoculation	Trial Conditions	References
** Competitive rhizosphere colonization + Antibiosis **					
Production of siderophores and antibiotics along with successful rhizotic zone colonization by these strains	Maize (*Zea mays*)	Root diseases (*Fusarium* *moniliforme*, *F. graminearum* and *Macrophomina phaseolina*)	*Bacillus* sp. MR-11(2) + *Bacillus* sp. MRF; *Pseudomonas fluorescens* sp. M23 + *Bacillus* sp. MRF	Pot	Pal et al., 2001 [33]
** Antibiosis + Enzyme lysis **					
Production of DAPG by *P*. *fluorescens* F113; production of extracellular proteolytic enzymes by *S. maltophilia* W81	Sugar beet (*Beta vulgaris*)	Damping-off (*Pythium* spp.)	*Pseudomonas fluorescens* F113 + *Stenotrophomonas maltophilia* W81	Pot and field	Dunne et al., 1998 [132]
** Competitive rhizosphere colonization + Induction of systemic resistance **					
Pseudobactin-mediated competition for iron for *P. putida* WCS358; induction of systemic resistance for RE8	Radish (*Raphanus sativus*)	Fusarium wilt (*Fusarium* *oxysporum* f. sp. *raphani*)	*Pseudomonas putida* WCS358 + *P. putida* RE8	Pot	de Boer et al., 2003 [135]
** Competitive rhizosphere colonization + Enzyme lysis **					
Production of siderophore as well as chitinase and β-1,3-glucanase by both strains	Fenugreek (*Trigonella foenum-graecum*)	Fusarium wilt (*Fusarium* *oxysporum*)	*Burkholderia* sp. RHT8 + *Burkholderia* sp. RTH12	*In vitro* and field	Kumar et al., 2017 [136]
** Competitive rhizosphere colonization + Antibiosis + Enzyme lysis **					
Siderophore production by *R. leguminosarum* RPN5; production of siderophore, HCN, chitinase, β-1,3-glucanase, β-1,4-glucanase by *Bacillus* sp. BPR7 and *Pseudomonas* sp. PPR8	Common bean (*Phaseolus vulgaris*)	*Macrophomina phaseolina*, *Fusarium oxysporum*, *F*. *solani*, *Rhizoctonia solani*, *Colletotrichum* sp. and *Sclerotinia sclerotiorum*	*Rhizobium leguminosarum* RPN5 + *Bacillus* sp. BPR7 + *Pseudomonas* sp. PPR8	Pot and field	Kumar et al., 2016 [137]
** Induction of systemic resistance **					
Induced systemic resistance by individual strains and their mixture	Mung bean (*Vigna* *radiata*)	Root rot and leaf blight (*Macrophomina phaseolina*)	*Pseudomonas putida* CRN-09 + *Bacillus subtilis* CRN-16	Pot	Sharma et al., 2018 [110]
Induced systemic resistance by *B. cereus* BS03 or *P. aeruginosa* RRLJ04, and the respective strain mixture	Pigeon pea (*Cajanus* *cajan*)	Fusarial wilt (*Fusarium udum*)	*Bacillus cereus* BS03 + *Rhizobium* sp. RH2; *Pseudomonas aeruginosa* RRLJ04 + *Rhizobium* sp. RH2	Pot	Dutta et al., 2008 [111]
Induced systemic resistance by a mixture of individual strains	*Arabidopsis thaliana*	Downy mildew (*Hyaloperonospora arabidopsidis*)	*Xanthomonas* sp. + *Stenotrophomonas* sp. + *Microbacterium* sp.	Pot	Berendsen et al., 2018 [112]
Induced systemic resistance by individual strains or their mixture	Tomato (*Lycopersicon esculentum*), pepper (*Capsicum annuum*) and cucumber (*Cucumis sativus*)	Southern blight (*Sclerotium rolfsii*) of tomato, anthracnose (*Colletotrichum gloeosporioides*) of pepper, and mosaic disease (*Cucumber mosaic virus*) of cucumber	*Bacillus amyloliquefaciens* IN937a + *B*. *pumilus* IN937b	Field	Jetiyanon et al., 2003 [133]
Increased superoxide dismutase (SOD) and peroxidase (PO) activities due to systemic resistance induced by the *Bacillus* strain mixture	Tomato and pepper	*Sclerotium rolfsii* and *Ralstonia solanacearum* in tomato; *S*. *rolfsii* and *Colletotrichum* *gloeosporioides* in pepper	*Bacillus amyloliquefaciens* IN937a + *B*. *pumilus* IN937b	Pot	Jetiyanon 2007 [134]
Induced systemic resistance by individual strains and their mixture	Rice (*Oryza* *sativa*)	Rice blast (*Pyricularia* *oryzae)*	*Pseudomonas fluorescens* Aur6 + *Chryseobacterium balustinum* Aur9	Field	Lucas et al., 2009 [138]
An increase in the enzyme activity including chitinase, β-1,3-glucanase, and polyphenol oxidase induced by both strains	Ginger (*Zingiber officinale*)	Rhizome rot (*Fusarium solani* and *F. oxysporum*)	*Bacillus subtilis* + *Burkholderia cepacia*	Pot and field	Shanmugam et al., 2013 [139]
** Signal interference **					
Degrading AHL by acylase and inhibiting biofilm formation		*Ralstonia* *Solanacearum* (single inoculation)	*Pseudomonas aeruginosa* 2apa	*In vivo*	Jayanna and Umesha, 2017 [120]

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
