# Peer review of "Pathogen Biocontrol Using Plant Growth-Promoting Bacteria (PGPR): Role of Bacterial Diversity"

_microorganisms, 2021, doi:10.3390/microorganisms9091988_

Round 1

Reviewer 1 Report

General comments

The manuscript represents a review of Plant Growth Promoting Rhizobacteria and their ability to suppress plant-pathogens and promote growth.  The authors attempt to define there is a need for PGPRs and then the different mechanisms, by which they suppress plant-pathogens.  The authors, then use literature case studies to highlight when combinations of PGPRs have been used to gain greater pathogen control and when there is a detrimental effect from the combinations of PGPRs.  The authors discuss the benefits and approaches required to improve the efficacy of PGPRs for controlling pathogens.  The authors conclude that combinations of PGPRs are better than use of single isolates. 

The manuscript requires extensive review to improve the language. There are long sentences, which do not make sense throughout the manuscript, but particularly in the abstract and conclusion.  The conclusion requires extensive revision, as it introduces new concepts not included in the body of the manuscript.  The manuscript develops some interesting concepts, such as synergistic, additive and antagonistic combinations of PGPRs.  However. it misses the opportunity to dig deeper to the understanding why some combinations are more effective than others and why some combinations do not work.  The authors should consider developing a greater understanding of the mechanism combinations listed in Table 1 and categorising the mechanisms.

Overall, the manuscript develops some interesting themes and ideas about PGPRs but could go a lot further into developing a better understanding of how to use PGPR to reduce plant-pathogens.

Specific section comments

Abstract

P1 “However, a cumulative or synergistic effect does not always ensue when bacteria introduced as a small mixture and interact with host plants” – This sentence does not make sense and requires fixing.

P1 “We also provide evidence of benefits when PGPR mixtures, rather than individuals, are used for protecting crops from various diseases, and underscore the critical determinant factors across different strains/species of PGPR in mixtures that offer greatest potential for successful biocontrol”. This sentence does not make sense and requires fixing.

P1 “Then we evaluate the challenges of and limitations to achieving desired out-comes from strain/species-rich bacterial assemblages, particularly in relation to their role in developing effective PGPR mixtures for plant disease management.” Delete superfluous text.

There is an average of 31 words per sentence in the abstract.  This makes reading extremely difficult.  The last sentence is a whopping 50 words long! The authors should be aiming for less than 25 words per sentence.  Therefore, the abstract requires rewriting to improve the clarity. Aim is particularly needed to write concise, shorter sentences.

Introduction

The problem and context of the problem is not clearly defined.  More emphasis is required on understanding the importance of plant pathogens and why the use of PGPRs is required to alleviate that problem.

P1 “In the rhizosphere micro-habitat plants continuously” Comma required after plants.

P2 “Since PGPR survive on the root surface or sometimes inside the root (i.e., as endophytes) as individual communities in natural soil and as are recruited by host plants, it has been deemed that each bacterial ‘component’ facilitates plant growth and health from specific plant-beneficial traits” – This sentence does not make sense and requires fixing.

Mechanisms of PGPR in disease suppression

Fig 1. What does the “Climate change” label infer? Is the climate change label needed in the figure?

P1 “colonize the root system, and which is well-recognised” – delete

P1 “microorganisms, including…” – add comma

P1 “root exudates consist of, among others, sugars….” Delete

The competitive rhizosphere colonisation mechanisms could be better defined in the paragraphs.  The manuscript alludes to three mechanisms,

  1. competition for carbon, 2.
  2. competition for nutrients and
  3. motility

Antibiosis

What examples are there of mixtures of PGPRs with different antibiotic production having a synergistic, additive or antagonistic effect?  This is suggested in the text but there is no supporting evidence that its mixtures give greater efficacy by producing a range of antibiotic products?

Enzyme lysis

P1 “Fusarium oxysporum f. sp. cucumerinum” should be italicised in the text

P2 “Importantly, it is highly likely that a wider array of enzymes produced by different species of PGPR in mixture will have greater ad-vantage of suppressing multiple pathogens present in the host rhizosphere due to complementary action of their lytic enzymes.”  What evidence is there to support this statement or is it only the authors opinion?

Activation of systemic resistance

Nice description of mechanism and benefits of mixtures.

Signal interference

P1 “….hydrolyze the AHL lactone ring [113],” – remove italics

PGPR mixtures in disease suppression

P1 “RHT8 and RTH12 both showed production of siderophores as well as chitinase and β-1,3-glucanase, the bacterial consortia led to aggressive colonization of fenugreek roots and suppression of Fusarium oxysporum along with increased growth and yield of fenugreek in both in vitro and field conditions, as compared to single inoculation and non-inoculated control [136].” – This sentence does not make sense and requires fixing. It is 52 word long.

What is the best combination of mechanisms between mixtures of PGPRs?  What proportional increase do mixtures confer over single isolates in the studies cited.

Table 1

There appear to be formatting issues with the table?

Mechanisms of action should match mechanisms listed in part 2 Mechanisms of PGPR in disease control e.g. Competitive rhizosphere colonization, Antibiosis, Enzyme lysis, Activation of systemic resistance and Signal interference. 

List the different mechanisms under each category e.g.

Competitive rhizosphere colonization

……

Antibiosis

DPAG

…….

Enzyme lysis

b-1,3-glucanase

b-1,4-glucanase

Chitinase

Activation of systemic resistance

….

Signal interference

……

Are the mixtures giving effects that are synergistic, additive or antagonistic? 

What is the proportional benefit of the mixtures?

Limitations to PGPR mixture

P2 “Secondary metabolites secreted by one organism that impose negative implications on reduced the growth of or the exertion of disease control from the other organism is another form of antagonism that can occur between two populations of biocontrol agents [145].” This sentence requires rewording.  A suggestion has been provided above.

Conclusion

This section requires re-writing.  It is difficult to understand the statement from the work.

From my understanding the authors are firstly highlight the mechanisms that PGPRs have on pathogens and plants, secondly how a combination can be beneficial and thirdly that some combinations are deleterious.  Therefore, the authors should highlight to best combination of attributes and the combination of attributes that should be avoided.  Furthermore what is then to additional advantage.

P1 Why have the authors introduced the term “micrbiocides” (spelt incorrectly in the text) in the conclusion?  This term has not been used elsewhere.  Use of terminology should be consistent.

P1 What is the difference between plant fitness and yield? Increased yield is an outcome from improved plant fitness.

What evidence is there those mixtures of PGPRs are better than single isolates?

The review needs to build on the previous descriptors developed in the manuscript.

The conclusion adds a number of new ideas into the manuscript.  The conclusion should summarise the key findings

Author Response

Thank Reviewer 1 for the detailed comments and suggestions for improvements. All the comments are very valuable and have been considered for revision. Please kindly find the attached point-to-point response to comments.

Reviewer 2 Report

Manuscript entitled “Pathogen biocontrol using plant growth-promoting bacteria (PGPR): role of bacterial diversity” fits within the journal scope. The topic is well defined and arouses the reader's interest.

Microbial communities inhabit the host plants focus the special attention of researchers. It should be remembered that in such multispecies communities of microorganisms the positive effect is the result of the interaction of many mechanisms promoting plant growth (positive and negative interactions). The Authors analyze and provide many examples of PGPR and mechanisms of diseases suppression. The Authors describe: (1) competitive rhizosphere colonization, (2) antibiosis and enzyme lysis, (3) activation of systemic resistance and  interference of signals. The Authors analyze the the role of the PGPR mixtures in disease suppression, and limitations that may occur in the potential consortia of PGPR .

Title is informative. Keywords are adequate. Abstract is concise and comprehensive. It summarize the most important conclusions and opinions of the Authors.The manuscript is divided into clearly defined sections. It contains introduction, detailed literature review divided into three chapters, and the conclusion.  Presented the literature survey is interesting and cited results are appropriately interpreted. The Authors’ considerations are very interesting and base on the latest data. Conclusions are supported by appropriate examples and correctly justifiable. Manuscript contains details allowing the readers understand the main matters. It is written attentively. According to me, this manuscript is in good relation to the latest research trends and specialist literature. Authors cited very interesting publications, and references list contains 163 articles.

Details that I would change:

  1. Page 5 – “of other compounds  like proteases [90]”  change to “of other enzymes like proteases”
  2. Page 6 – “Induced systemic resistance (ISR) is a state of active resistance due to an inducing agent after pathogen infection. ISR can be induced by beneficial rhizobacteria, whereas the pathogen-induced resistance is called systemic acquired resistance (SAR) [92].” According to me, the differences between ISR and SAR should be better described.
  3. Page 7 – “leading to more consistent broad-spectrum pathogen protection in various crops under field conditions in comparison with an individual strain” What did the authors mean – pathogen protection or pathogen control?
  4. Table 1 – In my opinion, Table 1 should be rearranged. The columns in this table are very narrow, which makes it difficult to read
  5. I also think that chapter 5 entitled "Concluding remarks" should be shortened.

In my opinion this manuscript will be attractive for a wide readership. Publishing this work will be benefit. I recommend this manuscript for printing after minor revision.

Author Response

We appreciate your time and valuable comments.

We carefully revised the manuscript, and addressed all comments. Please kindly find the attached “Authors’ Response to Comments” with point-to-point responses to each comment and suggestion with the amendments made to the manuscript.
